

# Role of mean and variability change for changes in European annual and seasonal extreme precipitation events

Raul R. Wood[1]

[1]Department of Geography, Ludwig-Maximilians Universität München, Munich, 80333, Germany

*Correspondence to*: Raul R. Wood (raul.wood@lmu.de)

**Abstract.** The frequency of precipitation extremes is set to change in response to a warming climate. Thereby, the change in precipitation extreme event occurrence is influenced by both a shift in the mean and a change in variability. How large the individual contributions from either of them (mean or variability) to the change in precipitation extremes are, is largely unknown. This is however relevant for a better understanding of how and why climate extremes change. For this study, two

sets of forcing experiments from the regional CRCM5 initial-condition large ensemble are used. A set of 50 members with historical and RCP8.5 forcing as well as a 35-member (700 year) ensemble of pre-industrial natural forcing. The concept of the probability risk ratio is used to partition the change in extreme event occurrence into contributions from a change in mean climate or a change in variability. The results show that the contributions from a change in variability are in parts equally important to changes in the mean, and can even exceed them. The level of contributions shows high spatial variation

which underlines the importance of regional processes for changes in extremes. While over Scandinavia or Mid-Europe the mean influences the increase in extremes more, reversely the increase is driven by changes in variability over France, the Iberian Peninsula, and the Mediterranean. For annual extremes the differences between the ratios of contribution of mean and variability are smaller, while on seasonal scales the difference in contributions becomes larger. In winter (DJF) the mean contributes more to an increase in extreme events, while in summer (JJA) the change in variability drives the change in

extremes. The level of temporal aggregation (3h, 24h, 72h) has only a small influence on annual and winterly extremes, while in summer the contribution from variability can increase with longer durations. The level of extremeness for the event definition generally increases the role of variability. These results highlight the need for a better understanding of changes in climate variability to better understand the mechanisms behind changes in climate extremes.

## 1 Introduction

Climate extremes (i.e., droughts, heat waves and floods) are set to change in a warming climate (Böhnisch et al., 2021; Brunner et al., 2021; Suarez-Gutierrez et al., 2020; van der Wiel et al., 2022) and recent devastating extreme events are testing the resilience of society. The rapid attribution of recent devastating extremes, such as the July 2021 Flood in Western Germany (Kreienkamp et al., 2021) or the heat wave in British Columbia (Philip et al., 2021) emphasize an already quantifiable influence of climate change on the severity of these and other extreme events. In observational records





significant trends emerge for various extreme metrics (Contractor et al., 2021; Fischer and Knutti, 2016; Fowler et al., 2021; Guerreiro et al., 2018; Westra et al., 2013). The impact of a warming climate on future precipitation extremes is a well-studied research field (Martel et al., 2021) with a consensus that precipitation extremes are increasing in magnitude and frequency over most parts of the world. Over Europe, it is shown that the magnitude (i.e., mean state) of extreme or heavy precipitation is increasing in Central and Northern Europe in all seasons while the Mediterranean Region can show

decreasing trends in summer (Aalbers et al., 2018; Hodnebrog et al., 2019; Poschlod and Ludwig, 2021; Rajczak and Schär, 2017; Rutgersson et al., 2022; Wood and Ludwig, 2020). At sub-daily timescales precipitation extremes can increase at higher rates then on daily timescales (Wood and Ludwig, 2020; Fowler et al., 2021). The general assumption is that the magnitude of precipitation extremes is likely to increase under a warming climate due to atmospheric warming and its inherent impact on the hydrological cycle (Allen and Ingram, 2002; Held and Soden, 2006). While mean precipitation is

constrained by the Earth's energy budget and scales at 1-3%/K per degree of global surface temperature warming, extremes are not constrained and can scale at the rate of moisture change at around 6-7%/K (O'Gorman and Schneider, 2009). Regionally and seasonally it is shown that precipitation extremes can considerably deviate from these global scaling rates, by scaling at levels well above the 7%/K Clausius-Clapeyron scaling (Wood and Ludwig, 2020; Lenderink et al., 2017; Poschlod and Ludwig, 2021; Lenderink and van Meijgaard, 2008) or showing negative scaling rates for seasonal extremes in

the Mediterranean (Wood and Ludwig, 2020; Bador and Alexander, 2022). The regional and seasonal response of extreme precipitation to global warming is thereby governed by thermodynamic and dynamic drivers (Brogli et al., 2019; Kröner et al., 2017; Pfahl et al., 2017; Norris et al., 2019; Vries et al., 2022).

Besides the change in the magnitude of extreme precipitation, the extreme event occurrence (i.e., frequency) is as well set to change under global warming (Martel et al., 2020; Myhre et al., 2019). Any changes to the distribution of precipitation,

hence also extreme events at the tail of the distribution, are influenced by both a shift in the mean and a change in variability. Thereby, the changes in the mean and variability can have different driving mechanisms (Pendergrass et al., 2017; van der Wiel and Bintanja, 2021; Bintanja and Selten, 2014; Bintanja et al., 2020). The variability connects the swings between extreme climatic states (Swain et al., 2018) and even when taking an evolving mean climate into account the change in variability influences the occurrence of extremes (Suarez-Gutierrez et al., 2020). Precipitation variability has been shown to

increase at a higher rate than mean precipitation with regionally diverse patterns (Pendergrass et al., 2017; Wood et al., 2021). In global climate model simulations, van der Wiel and Bintanja (2021) show that the contributions of climate variability to the change in monthly extreme precipitation is considerable and that the contribution shows strong regional variations. However, to analyze the contributions on the European scale, higher resolution regional climate simulations are required. Higher resolution regional climate models yield lower biases and show added-value in representing local climate

(Prein et al., 2016; Poschlod, 2021).

Extreme events with its rare occurrence are the most discernible manifestation of internal climate variability and more broadly precipitation projections are strongly influenced by the uncertainty of internal climate variability even far into the future (Lehner et al., 2020), especially on regional scales (Lehner et al., 2020; Wood and Ludwig, 2020). Hence, climate



simulations from a regional single model initial-condition large ensemble (SMILE) are used for a more robust sampling of

extreme events under pre-industrial, current, and future climate conditions. The benefit of using SMILEs for the robust
quantification of extreme event metrics has been asserted in many studies for numerous types of extremes. For example, the
added-value of SMILEs for a better quantification of rare flood events (van der Wiel et al., 2019; Brunner et al., 2021;
Kelder et al., 2022), the change in magnitude and frequency of precipitation extremes (Aalbers et al., 2018; Hodnebrog et al.,
2019; Martel et al., 2020; Poschlod and Ludwig, 2021; Wood and Ludwig, 2020; Thompson et al., 2017), or droughts

(Aalbers et al., 2022; Böhnisch et al., 2021; van der Wiel et al., 2022). SMILEs are also beneficial for studying changes in
precipitation variability (e.g., Maher et al., 2021b; Pendergrass et al., 2017; Wood et al., 2021), the changes in the driving
modes of climate variability (e.g., ENSO or NAO; Maher et al., 2018; McKenna and Maycock, 2021), and the robust
quantification of changes in weather patterns (Mittermeier et al., 2019; Mittermeier et al., 2022). An overview of other
applications using SMILEs can be found in Deser et al. (2020) and Maher et al. (2021a).

Here the probability risk ratio is used in regional large-ensemble climate simulations to partition the changes in extreme
annual and seasonal precipitation events into contributions from changes in mean climate and climate variability. It is further
analysed whether the contributions are influenced by the warming level, season, level of extremeness, or level of temporal
aggregation (3h-72h).

## 2 Data and Methods

### 2.1 Climate simulations

For this study, two sets of forcing experiments (ALL and NAT) with the Canadian Regional Climate Model version 5
(CRCM5) are used. The ALL forcing experiment originate from the CRCM5 large ensemble (CRCM5-LE; Leduc et al.,
2019). The CRCM5-LE is a regional 50 member initial-condition large ensemble, which was produced by dynamically
downscaling the 50 member CanESM2 large ensemble (Canadian Earth System Model version 2 large ensemble; Fyfe et al.,

2017; Kirchmeier-Young et al., 2017) with the regional climate model CRCM5 (v.3.3.3.1; Martynov et al., 2013; Šeparović
et al., 2013) to the EURO-CORDEX 0.11° grid in a one-way nesting setup. All 50-member use combined anthropogenic
(CO2 and non-CO2 GHGs, aerosols, and land cover) and natural (solar and volcanic influences) forcing (ALL forcings).
Historical forcing is applied before 2006, and RCP8.5 (Meinshausen et al., 2011) is used for 2006 until 2100. The
differences among the individual CRCM5 members are due to the macro and micro initialization in the driving CanESM2-

LE and can be interpreted as natural climate variability.

For the NAT forcing experiment, the CRCM5 uses the CanESM2 pre-industrial control simulations (Arora et al., 2011) as its
driving data. The pre-industrial simulations represent a climate state in 1850 without anthropogenic global warming at
constant atmospheric $CO_2$ levels of 284.7ppm. From this 1000-year CanESM2 pre-industrial continuous simulation, 35 non
overlapping time slices of each 22 years were selected and used as boundary conditions for the CRCM5 resulting in 35 pre-

industrial control members. From each of the 35 CRCM5 members, the first two years were discarded as spin-up, resulting





in an ensemble of 700 years (35 members x 20 years). The CRCM5 setup used for this pre-industrial ensemble is identical to the setup used in Leduc et al. (2019) for the CRCM5-LE. Both CRCM5 experiments share the same model parameterization of deep convection (Kain and Fritsch, 1990) and shallow convection (Kuo, 1965; Bélair et al., 2005) providing hourly precipitation outputs. At a resolution of 0.11° a discrete modelling of convection is not possible and needs to be parameterized within the regional climate model.

The CRCM5-LE precipitation data was evaluated in various studies, showing a good representation of the timing of maximum annual precipitation (Wood and Ludwig, 2020), as well as good agreement for ten-year return levels of 3h-24h annual maxima with observations (Poschlod et al., 2021) over Europe. The CRCM5-LE is further capable of simulating synoptic weather pattern (i.e., Vb-cyclone) which are relevant for long-lasting high impact rainfall events triggering floods in the Alpine Region (Mittermeier et al., 2019). Over Eastern North America, the CRCM5-LE also yields a good representation of the annual and daily cycle (Innocenti et al., 2019). An analysis of the general biases of the CRCM5 setup can be found in (Leduc et al., 2019). Future projections of the annual maximum precipitation in the CRCM5-LE over Europe show similar patterns and magnitudes to the 16-member EC-Earth-RACMO large ensemble (Aalbers et al., 2018; Wood and Ludwig, 2020). The CRCM5-LE also shows a comparable spread of internal variability to other regional SMILEs and a good agreement of interannual variability with observations (von Trentini et al., 2020). The good representation of interannual variability can also be asserted to the driving CanESM2-LE (Wood et al., 2021). The large-scale NAO teleconnections, which are relevant for the interannual to multi-annual variability over Europe, are properly propagated from the driving CanESM2-LE to the CRCM5-LE (Böhnisch et al., 2020). For the CanESM2 statistically robust NAO patterns have been evaluated under current climate conditions (Böhnisch et al., 2020).

## 2.2 Methods

Here the probability risk ratio is used in regional large-ensemble climate simulations to partition the changes in extreme annual and seasonal precipitation events into contributions from changes in mean climate and climate variability. The basis for the analysis is annual (seasonal) maximum precipitation, which is defined as the maximum precipitation sum within a season (DJF or JJA) and year. Precipitation sums are calculated with a rolling window of size 3h, 24h and 72h accounting for partial overlaps with preceding/trailing seasons (years) to receive the absolute annual (seasonal) maximum precipitation. Annual (seasonal) maxima are calculated for each ensemble member and grid cell separately.

### 2.2.1 Event probability

The probability risk ratio is a widely used metric in attribution studies (Kirchmeier-Young et al., 2019a; Kirchmeier-Young et al., 2019b; Otto et al., 2018b; Swain et al., 2020) to analyse the change in event probability. It requires event probabilities from two different climate simulations (Figure 1a), which are defined here as the number of annual (seasonal) maxima exceeding a local event threshold. The event threshold is valid for both simulations and is based on the NAT simulations,




calculated for each season separately. For the threshold definition, all 700 annual (seasonal) values are pooled and normalized by its mean (see Eq. 1).

$$RX_{norm} = (RX_i - RX_{NAT}) / RX_{NAT} \quad (Eq. 1)$$

Where $RX_{norm}$ is the normalised annual (seasonal) maximum precipitation, $RX_{NAT}$ the mean annual (seasonal) maximum precipitation in the NAT simulation, and $RX_i$ the values to be normalised. The normalization (Eq. 1) is valid for both NAT and ALL simulations by replacing $RX_i$ with ALL (NAT) values. A normalization is applied to receive a comparable threshold across the domain and season. Thresholds based on absolute values without a normalization can show high spatial and seasonal variability. After normalization the standard-deviation over all values is calculated and events exceeding two-

times (three-times) the standard-deviation are considered for the event probability (Figure 1a).

$$Threshold = N*std(RX_{norm, NAT}) \quad (Eq. 2)$$

Calculations of the threshold and event probabilities are performed for each grid cell separately. To ensure the same sample size in the NAT and ALL simulations, 35 random members have been picked from the full 50-member ALL simulations. The random sampling without replacement has no effect on the results and different sets of random samples will produce

only very small marginal differences.

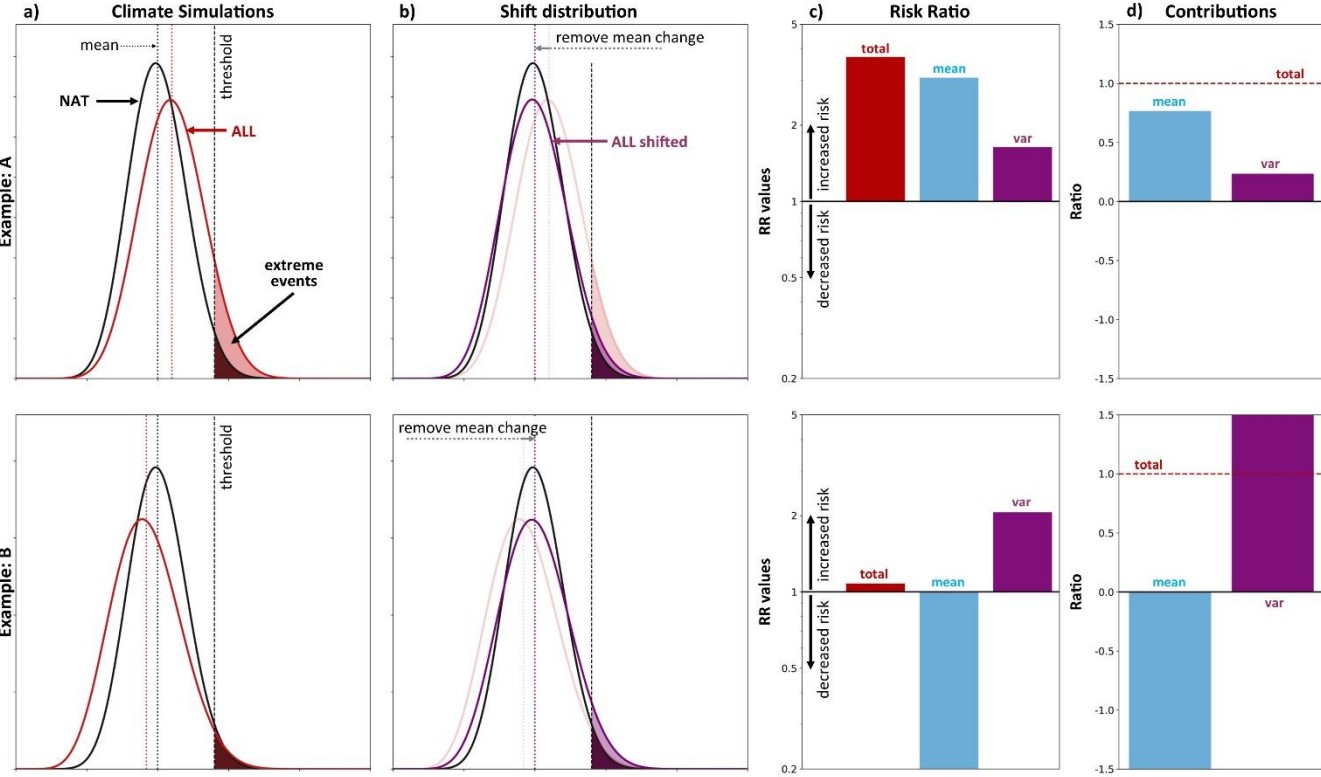

**Figure 1: Schematic of the probability risk ratio framework for separating contributions from mean and variability. Two examples are given. In example A both mean and variability contribute to an increase in event probability. Example B shows contrasting contributions from mean and variability. a) Shows two different climate simulations (NAT and ALL) for which the**



**PRtotal is calculated based on the number of events exceeding the threshold in both distributions. b) Any change in the mean is removed by shifting the ALL distribution to match the mean in the NAT distributions, the shifted ALL simulation is then used to determine the PRvar based on the events exceeding the threshold. c) The PRmean can be determined from an adapted probability risk ratio relationship, giving the PR-values for PRtotal, PRmean, and PRvar. d) The ratio of contribution is determined from the individual contributions from PRmean and PRvar to the PRtotal, which sum up to 1. For more details see the methods (section**
**2.2).**

### 2.2.2 Probability risk ratios

To assess the change in event probability, the framework of the probability risk ratio is applied. The conventional risk ratio as used in many attribution studies is calculated as follows:

$$PR = P_{ALL} / P_{NAT} \text{ (Eq. 3)}$$

with PR=1 indicating no change in extreme event probability, PR>1 indicates an increase in event probability and PR<1 a decrease in probability. Here, the event probabilities ($P_{ALL}$, $P_{NAT}$) are given as the number of extreme events in the ALL and NAT dataset and as described above. The conventional risk ratio framework is extended, as proposed by van der Wiel and Bintanja (2021) to separate the contributions from changes in the mean and changes in variability. The $PR_{Total}$ is calculated in the classical way by following Eq.3. The $PR_{Total}$ includes both the contributions from a change in the mean and variability

and therefore concludes the total change. To quantify the role of a change in variability (widening of the distribution), the influence of any change in the mean is first removed by shifting the entire distribution of ALL to match the mean of NAT (Figure 1b). The shifting is achieved by subtracting the difference in the mean of ALL and NAT. The shifting of the distribution is done prior to the normalization of the ALL precipitation values (i.e., Eq 1). The number of extreme events is determined in the new distribution and used to calculate the risk ratio $PR_{var}$, representing the change in event occurrence due

to the change in variability. From the two risk ratios $PR_{Total}$ and $PR_{var}$, the risk ratio for $PR_{mean}$ can be calculated following the new risk ratio relationship:

$$PR_{Total} = PR_{mean} + PR_{var} - 1 \text{ (Eq. 4)}$$

In this relationship subtracting by 1 is necessary because the reference PR-value is 1 (no change). The PR-values should be evaluated on a logarithmic scale, where PR=2 and PR=0.5 indicate a similar change in magnitude (Figure 1c).

### 2.2.3 Contributions from mean and variability

To quantify the relative contributions attributable to the change in the mean ($PR_{mean}$) and change in variability ($PR_{var}$) to the total risk change ($PR_{Total}$), a simple ratio of contribution is calculated as proposed by van der Wiel and Bintanja (2021):

$$C_{mean} = (PR_{mean} - 1) / (PR_{Total} - 1) \text{ (Eq.5)}$$

Which is equivalent for variability ($C_{var}$) by replacing $PR_{mean}$ with $PR_{var}$. The two contributions $C_{mean}$ and $C_{var}$ sum up to 1.

Thereby, they can either result in the same sign, which means that both mean and variability contribute to an increase (decrease) in the risk ratio (see Example A in Figure 1d), or they can have opposite signs showing opposing contributions (see Example B in Figure 1d). For the regional analysis the probability risk ratios (total, mean, and var) are averaged across





grid cells falling within the region boundaries (inclusion is based on cell centre points) before the ratio of contribution is calculated based on the regionally averaged PR-values.

### 2.2.4 Warming levels

Lastly, the risk ratios and their contributions are analysed for different warming levels. The warming levels are calculated from the driving CanESM2-LE dataset with a rolling window of 20 years with the pi-Control CanESM2 simulation as the reference. The ensemble mean warming is used to identify the 20-year periods closest to 1°C, 2°C, 3° and 4°C. Thereby, the 1°C warming level is considered as the current climate.

### 3 Results

#### 3.1 Probability risk ratio and ratio of contribution in annual extremes

#### 3.1.2 Current climate

Compared to a stable pre-industrial climate the present-day climate (+1°C) in the CRCM5-LE shows a widespread increase in the mean 3-hourly annual maximum (AX3h) precipitation by 4.6 % K$^{-1}$ over land (Figure 2a). The regionally averaged scaling rates differ between 3.6 and 5.9 % K$^{-1}$ among the different subregions. The standard deviation (i.e., variability) of the AX3h has increased by 8.9 % K$^{-1}$ over all land area within the same time (Figure 2b). The increase in variability is larger than the change in the mean AX3h in all subregions. The total probability risk ratio (PRtotal) of AX3h events larger than 2-sigma has also increased slightly by 1.36 averaged over all land areas (Figure 2e, Figure 4). This total change is influenced by both the change in the mean and variability. When the probability risk ratio is calculated based on the mean and variability separately, then slightly higher risk ratios can be seen for the PRvar (1.2) than for PRmean (1.16) (Figure 2c-d). The individual ratios of contribution for mean and variability to the total risk ratio show that the increase in the PRtotal can to a larger part be attributed to a change in variability (0.55 when averaged over all land area) and to a slightly lesser extend due to the mean (0.45) (Figure 2f-g, Figure 5). Within all subregions the contribution from variability varies between 0.48 and 0.63.

Other studies show that the observational records reveal an increased risk of extreme precipitation, at least when taking the change in mean extremes as a proxy (Westra et al., 2013; Westra et al., 2014; Donat et al., 2016; Sippel et al., 2017). Which in parts fits the trend seen in the CRCM5-LE, since the mean contributes to roughly 0.45 to the increase in extreme events. Although trends in single realizations (i.e., observations) could be underestimated since changes in variability are difficult to quantify from the limited sample size, studies from the detection and attribution community show that climate change is now detectable in everyday weather events (Sippel et al., 2020) and that recent extreme events over Europe have been amplified by climate change (Kreienkamp et al., 2021; Otto et al., 2018a), which makes the results from the CRCM5-LE for the seem plausible.





**Figure 2: Changes in the current climate (+1°C) compared to a stable pre-industrial climate in the CRCM5-LE simulations. a)**
**Change in the mean annual rx3h. b) Change in the variability (i.e., standard deviation of annual rx3h). c) PRmean, d) PRvar and**
**e) PRtotal values for 2-sigma events. f) ratio of contribution for changes in the mean. g) ratio of contribution for changes in**
**variability.**

### 3.1.2 Future climates

In a two-degree (+2°C) warmer world, the probability risk ratio continues to increase to 1.77 showing a doubling of 2-sigma
extreme events in roughly 29 percent of the land area (Figure 3a). The strongest increases in the total risk ratio can be seen in
the Scandinavian region with an average increase in the PRtotal by 2.1 with roughly 56 percent of grid cells showing a
doubling in events. By a change of mean or variability alone, a considerably smaller percentage of land area would show a
doubling in extreme events in Scandinavia (mean: 13 %, var: 6.3 %), and over all land areas (mean: 4 %, var: 3.5 %). This
emphasizes the joint role that changes in mean and variability have for shaping the total change in extremes. Both the
Scandinavian region and the Alps are clearly visible in the PRmean maps while the PRvar show a more widespread increase
in the risk ratio throughout the entire domain (Figure 3b-c).





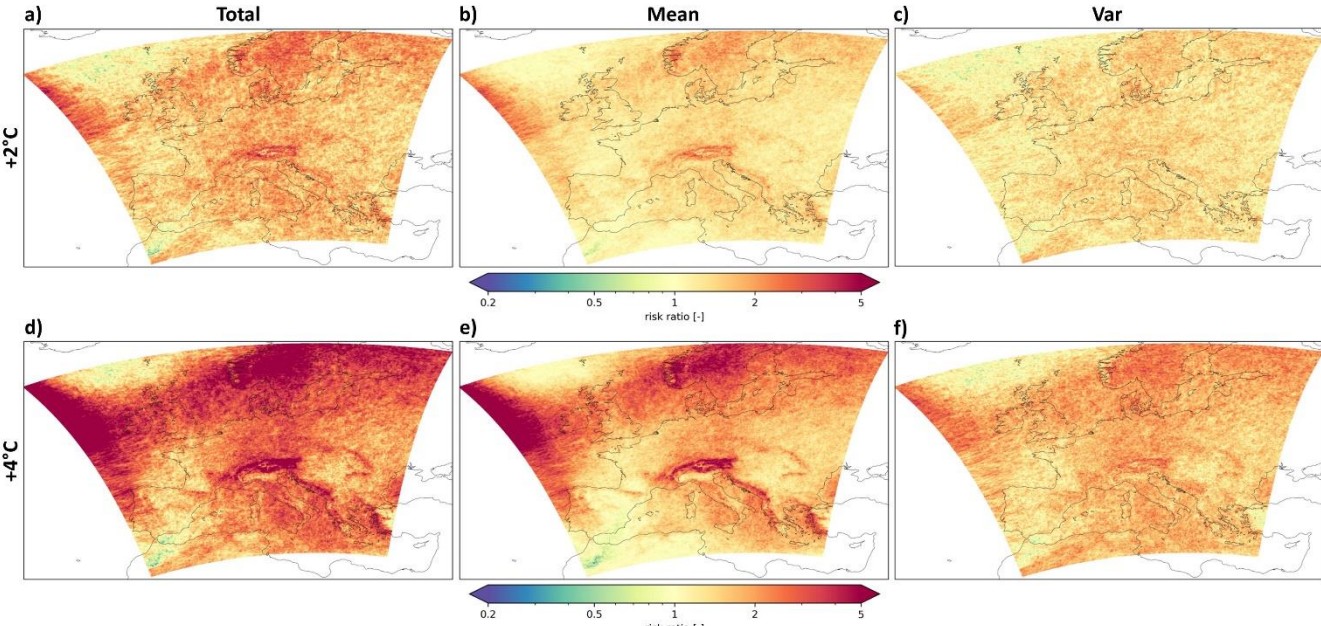

**Figure 3: Probability risk ratios for annual rx3h for extreme events larger than 2-sigma in a +2 °C and +4 °C warmer world. a) + d) PRtotal. b) + e) PRmean. c) + f) PRvar. a) – c) +2 °C climate. d) – f) +4 °C climate. All probabilities relative to the pre-industrial climate.**

In a four-degree (+4 °C) warmer world, the risk of 2-sigma extreme events becomes more likely with roughly 69 percent of land grid cells showing at least a doubling of events with an average increase in PRtotal of 2.7 (Figure 3d-f). While the PRvar is generally still increasing more widespread, the PRmean shows a more contrasting picture with regions, such as the Alps and Scandinavia, showing a very large increase in PR-values, while other regions show PR-values closer to one (i.e., no change), such as parts of the Iberian Peninsula or France. Figure 4 shows the regional average PR-values (total, mean, and var) for all PRUDENCE subregions at different warming levels, and reveals that in most regions the PRmean and PRvar develop similar. In Mid-Europe, Eastern Europe, and the Mediterranean both the PRmean and PRvar develop very closely and show almost identical PR-values. Over the British Isles the PRmean starts to increase steeper towards the +4°C warming level, diverging from the PRvar which shows a continued increase but at a lower level. In Scandinavia and the Alps, where the change in the PRtotal is most pronounced, the PRmean diverges already at +2°C from the PRvar and increases at considerably higher rates. Over France and the Iberian Peninsula, where overall PRtotal values are lower than in other regions, the PRvar remains throughout all warming levels slightly above the increase in PRmean. In all subdomains the probability of more extremes increases no matter if this is driven by a change in the mean or variability.





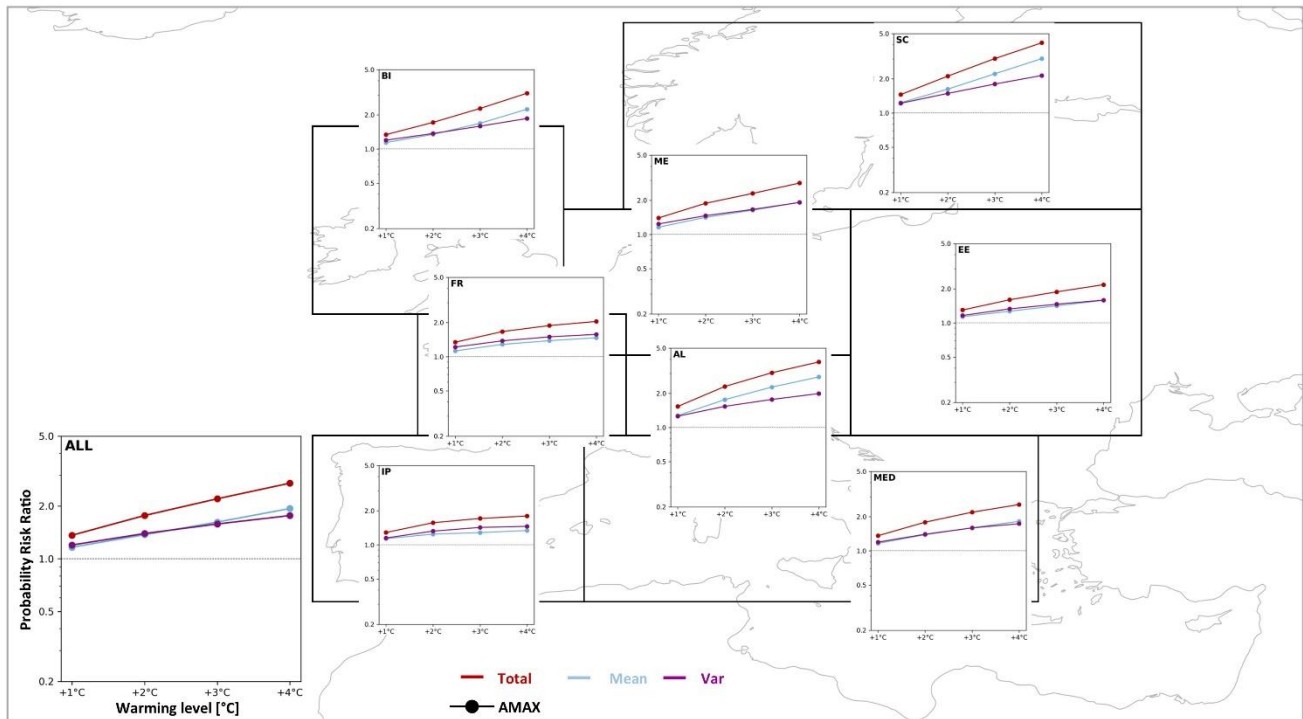

**Figure 4: Regional averaged PR-values (total, mean, and var) for the PRUDENCE regions at different warming levels for annual rx3h events larger than 2-sigma. PRtotal (red), PRmean (blue), and PRvar (purple) values (y-axis) at warming levels (+1, +2, +3, +4 °C) (x-axis). The lower left panel shows the aggregation over all land grid cells and shows axis labels.**

In Figure 5, the individual contributions from PRmean and PRvar to the total change (PRtotal) are shown for the subregions. Generalized over all land areas the contributions reveal that the change in variability attributes slightly more (approx. 0.55) in the current climate (+1°C) and the contributions steadily reduce to approx. 0.45 in the +4°C warmer world. This means the contributions from mean and variability develop diagonally to each other with the mean gaining in importance. On the regional scale however, there are distinct differences among the regions. The British Isles show a similar development to the domain average, but slightly more pronounced with the variability contributing by 0.58 in the current climate and by 0.41 at +4°C. In the Mediterranean this is less pronounced, and both contribute close to equally in the current and future climates. In Mid-Europe and Eastern Europe, the contributions from variability and mean converge with continued warming. In the current climate the variability has a higher contribution. Over Eastern Europe the converging takes slightly longer than over Mid-Europe where both (mean and variability) contribute equally from a +2°C climate onwards. In France, both contributions tend to converge, however the contributions from variability remain higher than the mean (0.55-0.63). In contrast, over Scandinavia and the Alps the contributions are approximately equal at current levels and diverge throughout the future warming with the mean gaining in importance (0.64 in both regions). Over the Iberian Peninsula the variability gains in importance towards a +3°C world (0.6) and slightly converges towards the end but remains higher than the mean.



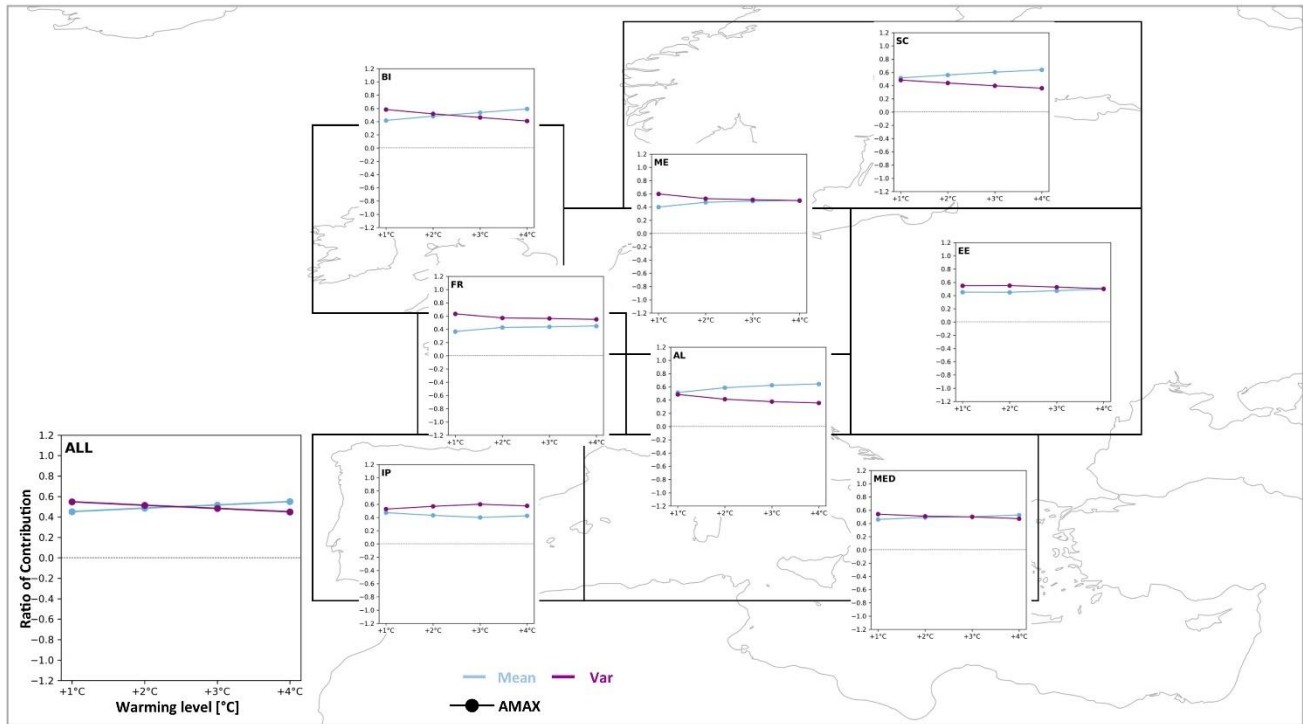

**Figure 5: Individual contributions from PRmean and PRvar to the PRtotal in the different PRUDENCE regions at different warming levels. Ratio of contributions from PR-values in Figure 4. Contribution from the mean in blue and contributions from variability in purple. Ratio of contribution on the y-axis and different warming levels on the x-axis. Warming levels: +1, +2, +3, +4 °C; The lower left panel shows the aggregation over all land grid cells and shows axis labels.**

## 3.2 Extremes on seasonal scales

### 3.2.1 Probability Risk Ratios

Looking at the seasonal scales which can be relevant for decision-makers the patterns reveal some interesting and diverse characteristics. Figure 6 shows maps of the probability risk ratios (PRtotal, PRmean, and PRvar) in the +4°C world for the two seasons winter (DJF) and summer (JJA) in comparison to the annual scale (as seen in Figure 3). The two seasons have been chosen since they show a strong seasonal contrast in the forced response of mean seasonal maximum precipitation as well as seasonal total precipitation amounts (Wood and Ludwig, 2020; Christensen et al., 2019; Matte et al., 2019; Rajczak et al., 2013).

In winter the increase in total risk ratio is in many parts of the domain larger than on the annual scales. Over Eastern Europe, the Greater Alps, the Balkan region as well as over the Iberian Peninsula more intense and widespread increases can be seen compared to the annual scale. Also, in winter the contrast between PRmean and PRvar is more pronounced with the mean projecting a higher probability of extremes. While the winter shows large widespread increases, in summer more grid cells





emerge that show a decrease, no change or only a marginal increase in the PRtotal. In general, the pattern of PRtotal follows the expected North-South gradient with increases in the north and decreases in the south. However, despite the summerly decrease in PRmean over France, Italy, Eastern Europe, the Balkan, and the Pyrenees, which clearly follows the decrease in the mean JJAx3h (see Figure S1 in the supplementary material), the PRtotal is still increasing in parts of these regions. Which means that the number of extremes is increasing even though the mean is decreasing and would project a decline in

extremes. Here, the decline in the risk ratio is compensated by the change in variability which is showing the opposite and shows an increase in the PRvar in these areas. This clearly highlights that the mean change is not always a sufficient proxy for the change in the probability of extremes. Especially over the Mediterranean and the Iberian Peninsula a widespread decline in the mean summerly average extremes is projected, however due to the change in the variability the probability of summerly extremes greater than 2-sigma remains and can even increase locally. Other clearly visible features in summer are

the Alps and Scandinavia, which are also apparent features in winter and on the annual scale.

**Figure 6: Annual probability risk ratios of rx3h events compared to seasonal DJF and JJA PR-values at +4 °C warming. a) – c) Annual PR-values; d) – f) DJF PR-values; g) – i) JJA PR-values; a) + d) + g) PRtotal; b) + e) + h) PRmean; c) + f) + i) PRvar.**



Through the regional aggregation some generalized statements can be formulated. Aggregated over all land areas, the PRtotal increase is strongest in DJF (3.34) compared to the annual scale (2.7) and lowest in JJA (2.06) at +4°C warming (Figure 7). Generally, this can also be shown for France (DJF: 2.8, AMAX: 2.04, JJA: 1.6), the Alps (DJF: 5.6, AMAX: 3.78, JJA: 3), and Eastern Europe (DJF: 4.18, AMAX: 2.17, JJA: 1.6). In these regions the PRtotal increases for the two seasons and the annual values. Also, the Iberian Peninsula and the Mediterranean show the same order of strongest to lowest increases, but with the unique characteristic that in JJA the PRtotal is decreasing in the Iberian Peninsula (0.71) and declining towards no change in the Mediterranean.

A different order can be seen over Scandinavia and Mid-Europe where the PRtotal in JJA and the annual scale are basically identical in their progression with warming. In Scandinavia, the PRtotal in DJF remains below JJA and the annual values for all warming levels. In Mid-Europe, values of JJA remain below DJF and the annual values until the +4°C world where all three values converge to approx. 2.7-2.8 (PRtotal). In the British Isles, the PRtotal is largest on the annual scale and is closely followed by JJA and shows a weaker increase in winter.

Generally, when comparing the evolution of PRmean and PRvar it can be stated, that in summer the PRvar is above the PRmean, and in winter vice versa. Except for Scandinavia where PRmean is always larger than PRvar. On annual scales, both the PRmean and PRvar are generally quite similar except for the Alps and Scandinavia where PRmean is considerably larger than PRvar.





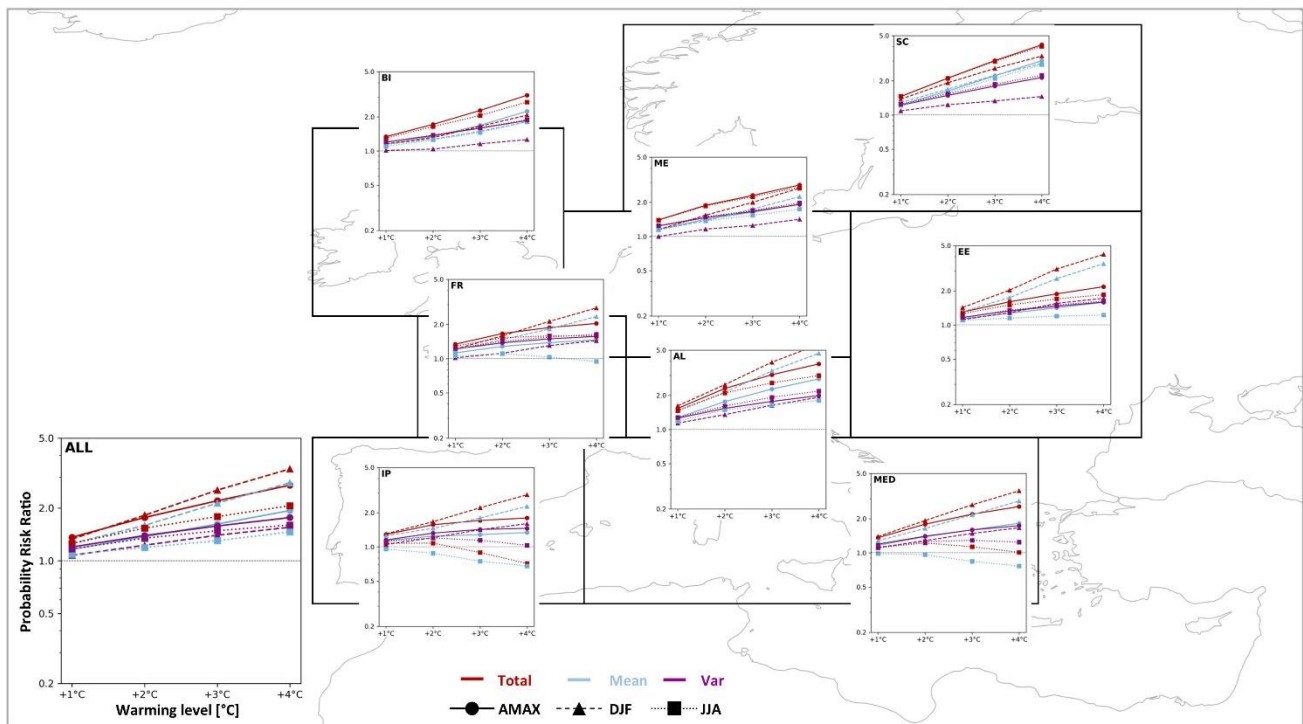


**Figure 7: Comparison of regional averaged annual and seasonal PR-values (total, mean, and var) at different warming levels. The panels show PRtotal (red), PRmean (blue), and PRvar (purple) values (y-axis) at warming levels (+1, +2, +3, +4 °C) (x-axis). The solid lines with the circle marker represent annual PR-values (same as in Figure 4); the dashed lines with the triangle marker represent PR-values in winter (DJF); the dotted lines with the square marker represent PR-values in summer (JJA). The lower left**
**panel shows the aggregation over all land grid cells and shows axis labels.**

### 3.2.2 Ratio of Contribution from mean and variability

In Figure 8 the ratios of contribution for JJA, DJF and the annual scale are compared. All regions, except for Scandinavia show the general behaviour that the variability contributes to a large extent to the change in extremes in summer, while in winter this relation is opposite (i.e., mean > var). Aggregated over all land areas, the variability attributes to 0.56-0.66 of the
change in summer while the mean only contributes to 0.34-0.44 of the change. In winter, the contribution of the variability only contributes to roughly a quarter (0.23-0.28) while the mean dominates the change in probability by roughly three-quarters (0.72-0.78). In comparison on the annual scale either the mean or variability contribute closer to equal by 0.45-0.55. Over the British Isles, the change in variability initially contributes to 0.7 (mean: 0.3) of the current change in the probability of summerly extremes before the contribution of both variability and mean converge to roughly equal contributions in a
+4°C world. In winter, the mean initially contributes to most of the change with roughly 0.9 (var: 0.1) and slowly reduces to 0.76 (var: 0.24).





Over the Alps the ratios of contribution are very stable across all warming levels within their respective season. In summer, the variability contributes to a higher degree with roughly 0.6 compared to 0.4 from the mean. In winter, the change in probability is dominated by the change in the mean contributing by 0.8 (var: 0.2).

Also, in Scandinavia the ratio of contribution remains very stable across the warming levels in winter, with the mean contributing roughly by 0.8 to the overall change (var: 0.2). In summer, both the mean and variability initially contribute almost equally to the change and diverge to roughly 0.6 attributable to the change in the mean compared to 0.4 by the variability.

Over Eastern Europe, the variability attributes roughly to 0.6 (mean: 0.4) of the current change in summer and increases to 330 0.7 (mean: 0.3) in future climates. In winter, the contributions are stable across warming levels and the mean attributes to roughly 0.75 (var: 0.25) of the change.

Over Mid-Europe, the difference in contributions between mean and variability is initially larger, and they slightly converge in a warmer climate. In summer, the variability contributes to 0.63 (mean: 0.37) of the total change before the two contributions converge slightly. In winter, the current change is predominantly driven by the change in the mean (close to 335 1.0) before the variability slightly gains in importance with roughly 0.25 (mean: 0.75) in warmer climates.

Over France, the ratios of contribution are experiencing considerable changes throughout the different warming levels and seasons. In winter, the mean contributes by 0.9 to the current change before reducing slightly to 0.75. In the same time contributions from variability increase from 0.1 to 0.25. In summer, the variability is the main driver of change with 0.8 at current climate levels and increasing beyond 1 in the future climate. A contribution beyond 1 is possible because the mean 340 contributes negatively to the change in the total risk ratio while variability shows an increase in extremes attributing to an overall increase in summerly extremes. This exemplifies that the change in the mean and variability can not only amplify the change in event probability, but in some cases counteract each other.

Over the Iberian Peninsula, the decline in the mean is responsible for the overall decline in the probability of extremes in summer. While the mean contributes to a decline throughout all warming levels, the variability can initially offset the overall 345 decline in summerly extremes but can't compensate for the strong decline in the mean in warmer climates. Note that the change in the sign of contributions in JJA is due to a change in the PRtotal shifting from an increase (>1) to a decrease (<1). However, locally in the northern parts of the Iberian Peninsula increases in the probability of extremes in summer can still occur due to the change in variability even though the mean is strongly decreasing (as seen in Figure 6). In winter, for which the PRtotal is continuously increasing, the mean contributes initially with 0.83 (var: 0.17) and is subsequently lower in 350 warmer climates (0.66-0.69).

Also, over the Mediterranean the mean contributes continuously to a decline in summerly extremes, however here the change in variability can initially offset the decline and lead to an increase in the probability of extremes in summer before the reversal of the trend towards no change of extremes in the +4°C world which is slowed by the presence of variability. In winter the mean attributes to roughly 0.7 of the change while variability by 0.3. The contributions are thereby stable across 355 all warming levels.





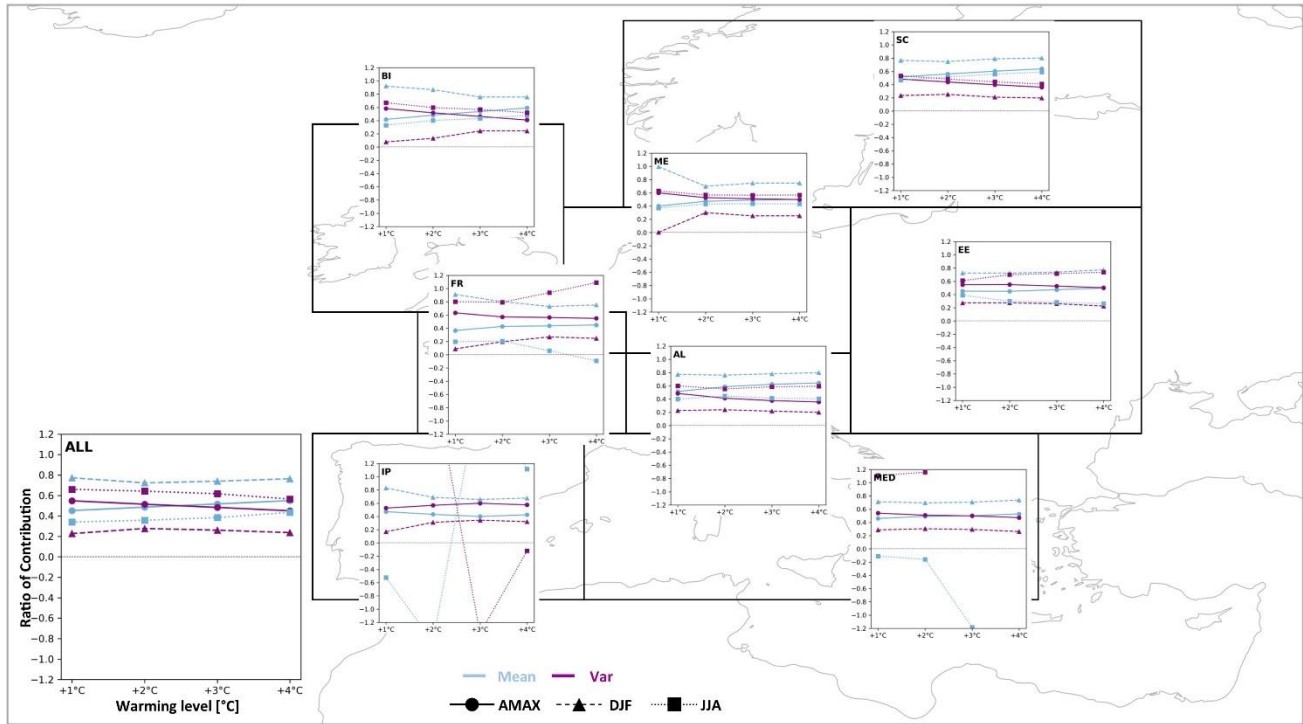

**Figure 8: Comparison of individual contributions of annual and seasonal PRmean and PRvar to the PRtotal at different warming levels. Ratio of contributions from PR-values in Figure 7. Contribution from the mean in blue and contributions from variability in purple. Ratio of contribution on the y-axis with different warming levels on the x-axis (+1, +2, +3, +4 °C). The solid lines with the circle marker represent annual ratio of contributions (same as in Figure 5); the dashed lines with the triangle marker represent ratios in winter (DJF); the dotted lines with the square marker represent ratios in summer (JJA). The lower left panel shows the aggregation over all land grid cells and shows axis labels.**

## 3.3 Influence of the temporal aggregation

Until now, all results shown are for an aggregation level of three hours raising the question whether the level of aggregation (i.e., 24-hours, 72-hours) has any influence on the ratio of contribution. First, looking at the probability risk ratios of annual extremes reveals that the level of temporal aggregation influences the magnitude of the probability risk ratios of total, mean and variability. In general, the PR-values of subdaily extremes (3-hours) are in most regions and aggregated over all land area higher than for 24-hours and 72-hours. Only over France the 3-hourly and 24-hourly PRtotal values develop close to identical with the 72-hours showing slightly lower values before all three aggregations converge in a similar PRtotal at +4°C. In Scandinavia, both the 24- and 72-hour extremes show near identical PR-values well below the 3-hour aggregation.



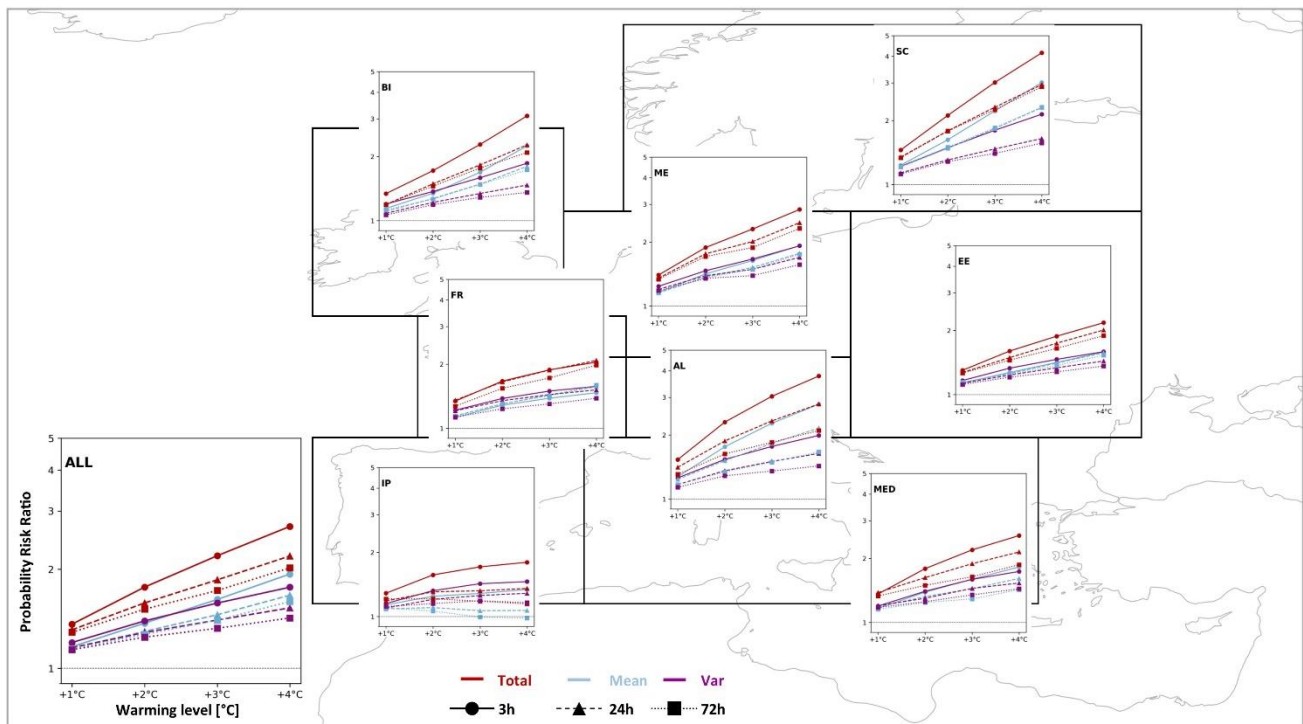

**Figure 9: Regional probability risk ratios for different temporal aggregation levels (3h, 24h, 72h) on annual scales. The panels show PRtotal (red), PRmean (blue), and PRvar (purple) values (y-axis) at warming levels (+1, +2, +3, +4 °C) (x-axis). The solid lines with the circle marker represent PR-values for 3-hour temporal aggregation (same as in Figure 4); the dashed lines with the triangle marker represent PR-values for 24-hours; the dotted lines with the square marker represent PR-values for 72-hours. The lower left panel shows the aggregation over all land grid cells and shows axis labels.**

The level of temporal aggregation has however only a very marginal influence on the ratio of contribution and the main takeaways from the previous sections remain true. Only in the Iberian Peninsula the influence of the variability considerably gains in importance. This is caused by a decrease in the PRmean in the 24-hour and 72-hour extremes. In the 3-hour data all PRtotal, PRmean, and PRvar show an increase, while in the 24h and 72h the PRmean shows a downward trend and in the 72h even a decrease in the PRmean from +3°C warming on. In comparison, the PRvar continues to increase in the 24h and increases then decreases in the 72h data.



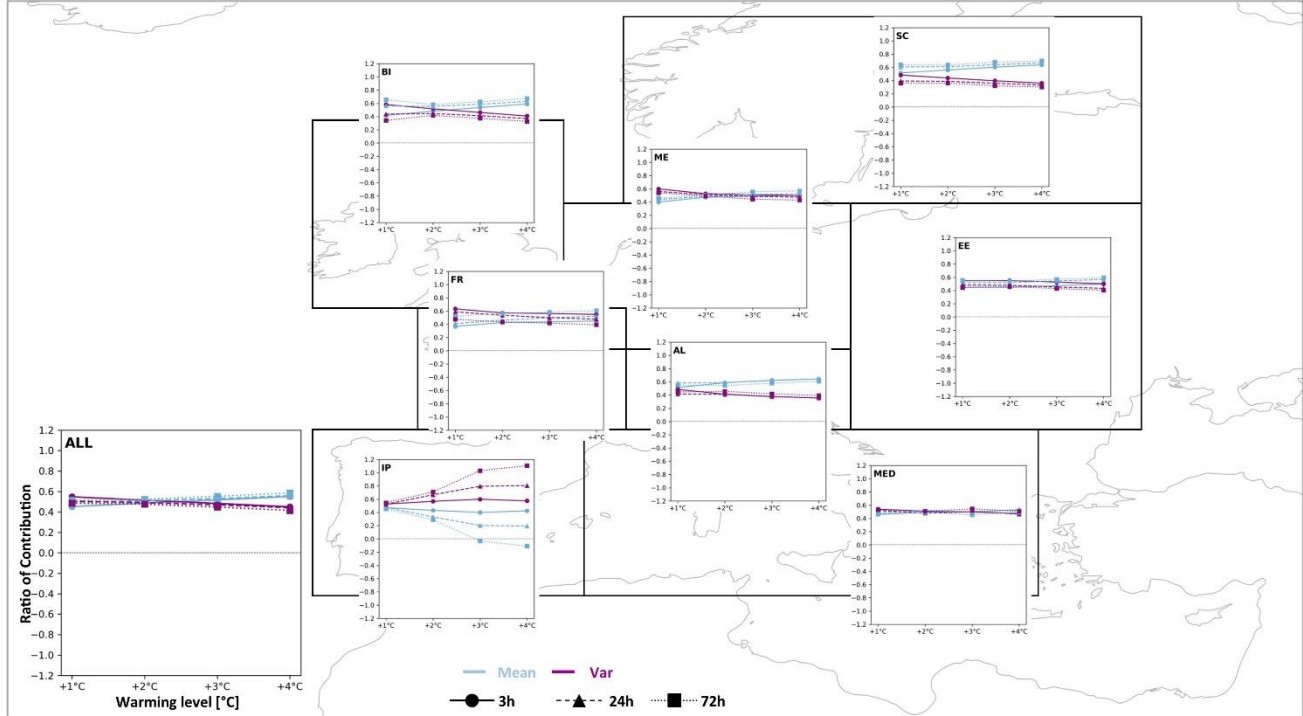

**Figure 10: Regional ratios of contribution based on different levels of temporal aggregation (3h, 24h, and 72h) for annual maxima. Ratio of contributions from PR-values in Figure 9. Contribution from the mean in blue and contributions from variability in purple. Ratio of contribution on the y-axis with different warming levels on the x-axis (+1, +2, +3, +4 °C). The solid lines with the circle marker represent individual contributions for 3-hour temporal aggregation (same as in Figure 5); the dashed lines with the triangle marker represent contributions for 24-hours; the dotted lines with the square marker represent contributions for 72-hours. The lower left panel shows the aggregation over all land grid cells and shows axis labels.**

Winter shows generally the same influence of temporal aggregation as seen on the annual scales. The PR-values are generally lower in the longer durations then in the subdaily extremes (Figure S2). In the British-Isles, Mid-Europe, Eastern Europe and over all land areas the three aggregation levels produce very similar PR-values throughout. Only in Scandinavia the longer durations show considerably higher PR-values then on the subdaily scale (PRtotal for 3h: 3.3, 24h: 4.2, 72h: 4.4). Over the Alps (PRtotal, 3h: 5.6, 24h: 3.6, 72h: 2.8) and the Iberian Peninsula (PRtotal, 3h: 2.9, 24h: 1.5, 72h: 1.3) the longer duration PR-values are markedly lower. Also, over France and the Mediterranean the PR-values are lower in the 24h and 72h data. However, these differences in the PR-values have only a low influence on the overall ratio of contributions which remain almost unaffected in the subregions of Scandinavia, Eastern Europe, the Alps, and the Mediterranean as well as aggregated over all land area (Figure S3). Over Mid-Europe the influence of the variability gains in importance for explaining the changes in the current (3h: ~0, 24/72h: ~0.3) and near-term future climate (3h: ~0.3, 24/72h: ~0.4). In the +3 and +4°C climates the ratios of contribution are near identical on all temporal aggregation levels. In the British-Isles the mean contributes more to the changes in the current climate in both the 24 and 72h data. In the future climates ratios are





similar across aggregation levels. In France, the variability in the 24-hours gains slightly in importance in the current climate

compared to the 3-hours. In the 72h data the mean gains in importance in current climate and slightly in future climates. In the Iberian Peninsula the 3h and 24h ratios are near identical, but in the 72h data the variability loses in importance especially in the +4°C climate due to the decrease in PRvar towards no change (1) from a previous increase (>1).

However, in summer the ratio of contribution is markedly influenced by the level of temporal aggregation (Figure S5). Aggregated over all land area this results in the variability contributing by 0.7-0.76 in the 24h data and 0.74-0.87 in the 72h

data compared to 0.56-0.66 in the 3h data. The gain in importance of the variability for changes in the probability of extremes with the level of aggregation can be seen in all regions. Differences due to the level of aggregation are less defined in the regions of Scandinavia and Mid-Europe, but very noticeable in France, the Alps, Eastern Europe, the Iberian Peninsula, and the Mediterranean. These differences in the ratio of contribution can be explained by the mean showing progressively decreasing PR-values (<1) or values closer to no-change with longer durations. The PRmean values of the 24h

and 72h are markedly lower than for the 3h data, while the temporal aggregation produces less of a difference in the PRvar values (Figure S4). As a result, the importance of the variability for the future changes in extreme event probability increases with temporal aggregation in summer.

### 3.4 Influence of the level of extremeness

The level of extremeness (2-sigma or 3-sigma) does in general not change the overall conclusions of the importance of both

the mean and variability for the total change in extreme events. The regions largely show the same order of importance by either the mean or variability. For example, regions where the mean contributes more to a change in event probability then the variability will also show this behaviour with a higher threshold for the event definition. However, the level of extremeness does in general increase the ratio of contribution for variability and respectively lowers the ratio of the mean. This increase in the ratio of contribution for variability is true for the annual scales (Figure S6) as well as the seasonal scales

(Figure S7, S8). Further, this can also be shown for the different temporal aggregations (Figure S9, S10). On the seasonal scale the order of contribution is unchanged with the mean showing higher contributions in winter, and the variability showing higher contributions in summer. On the annual scales where the ratios of contribution are relatively similar anyway the increase in the ratio for variability can change the major contributor from mean to variability. In regions where the mean and variability contributed near equal (e.g., Mid-Europe, Mediterranean) the contributions from the variability remain above

the mean with the 3-sigma threshold. Regions where the main contributor switched throughout continued warming from variability to mean (e.g., British-Isles, all land area) also show for the 3-sigma events that the contribution from variability remains larger than the mean, but the ratios converge to near equal in the +4°C world.



## 4 Discussion

In this study, only one regional large ensemble has been used which makes it difficult to evaluate the importance of model
uncertainty on these results. Using multiple global SMILES van der Wiel and Bintanja (2021) have shown that the model
uncertainty seems to only play a minor role for the contributions of mean and variability to the extreme event occurrence.
However, different models will influence the magnitude of the probability risk ratios. On the local scale different regional
climate models can show different land-atmosphere feedbacks, due to a difference in model components or parameterization,
which can influence the evolution of local precipitation extremes (e.g., Ritzhaupt and Maraun, 2023). Other regional
SMILES are necessary to analyse the impact of model uncertainty on the results. However, the availability of other regional
SMILES is limited. The only two other regional SMILES over Europe (to the knowledge of the author) differ in the extent of
the domain (Aalbers et al., 2018) or the model resolution (Brönnimann et al., 2018; Addor and Fischer, 2015). von Trentini
et al. (2020) have analyzed the three regional SMILES and show that the three SMILES reveal comparable changes in
interannual variability of various climate indicators. Comparing projections for seasonal maximum precipitation in the 50-
member CRCM5-LE (Wood and Ludwig, 2020) and the 16-member EC-Earth-RACMO ensemble (Aalbers et al., 2018)
reveals very comparable forced changes in the mean magnitudes. This might indicate that the findings in van der Wiel and
Bintanja (2021) of a small influence of model uncertainty on the ratio of contribution could potentially also be true for
regional SMILEs.

Over the Mediterranean region, including the Iberian Peninsula, it has been shown that the magnitude of the drying trend
especially for total summerly precipitation as well as mean extreme magnitudes can be model dependant, however there is a
high model agreement on an overall drying (e.g., Ritzhaupt and Maraun, 2023; Zittis et al., 2021). However, it has also been
shown that lower likelihood precipitation extremes still increase in the northern parts of the Mediterranean region (e.g., Zittis
et al., 2021). Both, the reduction in mean climate characteristics while upper tails increase, fit the results shown in this study
and strengthen the hypothesis that the increase in lower likelihood precipitation events is mainly driven by an increase in the
variability. Most regional climate simulations place the French domain within a transitional zone between a drying signal of
summerly precipitation in the south and a wetting in the north of Europe (e.g., Aalbers et al., 2018; Ritzhaupt and Maraun,
2023; Wood and Ludwig, 2020), largely showing no-change or a slight decrease in mean-state extremes, which is consistent
with the results here. This means that any increase in the upper tails is dependent on the change in variability.

Scenario uncertainty could also have an influence. However, by using warming levels instead of fixed time periods and
under the assumption that there is a physical basis for the connection of level of warming and climate system response, the
scenario uncertainty can be reduced at least for the warming levels which are reachable by both lower and higher emission
scenarios. To fully address the influence of scenario uncertainty on the presented results, a regional SMILE with multiple
dynamically downscaled emission scenarios from the same global model would be necessary. Unfortunately, such a multi
scenario regional SMILE ensemble does not exist.



Several studies have highlighted that convection permitting climate models (CPM) are better in representing precipitation extremes compared to regional climate models on non-convection resolving resolutions, especially in summer for convective events (e.g., Ban et al., 2014; Kendon et al., 2017; Pichelli et al., 2021). These studies are however often only a single model with a single short time slice simulation. Progress is being made on the availability of a multi-model CPM ensemble (Coppola et al., 2020; Pichelli et al., 2021). However, these simulations will only cover a small part of the Pan-European

domain and will rely on short time slice simulations of single climate realizations. These single decadal climate realizations will however be strongly influenced by natural climate variability (Lehner et al., 2020; Leduc et al., 2019; Deser et al., 2012; Hawkins and Sutton, 2009). Poschlod (2021) has shown the suitability of the CRCM5-LE and highlights the added value of using a regional SMILE for the analysis of precipitation extremes even on non-convection permitting resolutions. Other studies have shown that the CRCM5-LE, even though convection is parameterized, can show a good representation of the

timing of maximum annual precipitation (Wood and Ludwig, 2020), as well as good agreement for ten-year return levels of 3h-24h annual maxima with observations (Poschlod et al., 2021) over Europe. Concerning overall patterns of precipitation change in CPM compared to RCM ensembles, Pichelli et al. (2021) have shown that both ensembles are largely in agreement on the patterns of the change (over the Alps and northern Mediterranean) but that differences might occur in the magnitudes. This will likely entail that the magnitudes of the probability risk ratios will be different in the CPM models. However, this

does not necessarily mean a change in the relation between the influences of the mean and variability. The level of temporal aggregations or the level of extremeness also influence the magnitudes of the PR-values, but do not necessarily entail a change in the ratios of contribution. Further, Kendon et al. (2017) have shown that CPM and RCM simulations agree on many aspects of the change in future precipitation projections.

## 5 Conclusion

In this study, climate simulations from the regional CRCM5 initial-condition large ensemble are used to analyse the general drivers for the change in extreme annual and seasonal precipitation event probability. The concept of the probability risk ratio is used to partition the change in extreme event occurrence into individual contributions from a change in mean climate and a change in variability. The results reveal that for the increase in event probability of annual maxima larger than 2-sigma, both the change in the mean and variability contribute near equally to the total change. For seasonal extremes in

winter (DJF) the change in the mean is the major contributor to the total change. In summer the contribution from the change in variability is larger than the mean, and in some regions, variability is the sole driver of an increase in extreme event occurrence. Over France, the Iberian Peninsula, and the Mediterranean the change in variability can lead to an increase in extreme event probability despite a strong decline in extreme precipitation events as projected by the mean. The strong decrease in the mean would likely entail a decrease in the probability of extreme precipitation events, but due to an increase

in variability the overall probability can still increase or remain at current levels. The level of extremeness in the event definition (2-sigma or 3-sigma) does in general not change the overall results of this study. Also, the level of temporal



aggregation is generally not changing the results. However, both do tend to increase the importance of the variability slightly.

## Data availability

The CRCM5-LE data for the historical and RCP8.5 simulations are available through https://climex-data.srv.lrz.de/Public/. The CRCM5-LE pre-industrial control simulations are available upon reasonable request.

## Author contribution

RRW designed the study concept, performed all analysis including visualization of results, and wrote the manuscript.

## Competing interests

The author declares that there is no conflict of interest.

## Acknowledgments

CRCM5 was developed by the ESCER Centre of Université du Québec à Montréal (UQAM) in collaboration with Environment and Climate Change Canada. We acknowledge Environment and Climate Change Canada's Canadian Centre for Climate Modelling and Analysis for executing and making available the CanESM2 Large Ensemble simulations used in
this study, and the Canadian Sea Ice and Snow Evolution Network for proposing the simulations. Computations with CRCM5 for the ClimEx project were made on the SuperMUC supercomputer at the Leibniz Supercomputing Centre (LRZ) of the Bavarian Academy of Sciences and Humanities. The operation of this supercomputer is funded via the Gauss Centre for Supercomputing (GCS) by the German Federal Ministry of Education and Research and the Bavarian State Ministry of Education, Science and the Arts. The CRCM5-LE simulations used here were produced for the ClimEx project funded by the
Bavarian State Ministry of the Environment and Consumer Protection (grant no. 81-0270-024570/2015).

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
