# Peer review of "Role of mean and variability change for changes in European annual and seasonal extreme precipitation events"

_EGUsphere, 2023_

## Author Response (AR1)

The two minor revision requests from the two anonymous reviewers have been addressed. A brief list of the relevant changes can be found below followed by the detailed comments to the individual reviewer comments (same as in the interactive comment section) including the associated revisions taken (indicated by the green arrow).

**List of relevant changes:**

- Updated Figure 1: (Replaced „NAT" with „PIC" in the Figure)
- Updated Figures 4, 5, 7, 8, 10: Adjusted Y-Axis for better readability
- Added minor clarifications throughout the manuscript
- Added additional credits throughout the manuscript regarding the main method by van der Wiel and Bintanja (2021)
- Extended the discussion as suggested by reviewers

**Please see the Reviewer comments (black) and the detailed replies (in green) below. The revisions taken are indicated by the green arrow.**

*In this study, the author investigated the contributions of the changes in mean and variability of climate to the change in precipitation extreme events over Europe, basing on CRCM5 simulations. The author suggested that the contributions from a change in variability are in parts equally important to changes in the mean, and can even exceed them. For the level of contributions, there are regional differences. In addition, the contributions also show differences in summer and winter. The topic is interesting and the results are valuable to better understand the changes in the precipitation extremes over Europe. I have following comments and suggestions:*

1. *I remember that the pre-industrial control simulations did not include the natural forcing. If so, the CRCM5 simulations drived by the CanESM2 pre-industrial control simulations cannot be used as the NAT forcing experiment.*
   Thank you for this comment. To avoid future confusion the NAT forcing experiment will be renamed to PIC (pre-industrial control). The relevant information on forcing is already included in the manuscript. The PIC simulation uses constant atmospheric $CO_2$ concentrations of 284.7 ppm. The concept of this work is unaltered by the change in naming, since the goal is to compare a world with climate change (RCP8.5) with a world before anthropogenic climate change (pre-industrial).
   ➔ **Changed the naming of the baseline climate simulation "NAT" with "PIC" throughout the manuscript as well as in Figure 1.**

2. *Are the warming levels of 1, 2, 3, and 4 Celsius degrees calculated from global mean or regional mean? The related information should be given in the manuscript.*
   The warming levels have been derived from the driving CanESM2 model and are calculated as Global mean temperature changes. This will be clarified in the manuscript.
   ➔ **Clarified within the manuscript now.**

3. *In this study, the author used the 50-memer initial-condition large ensemble. So the result uncertainty from the initial conditions should be given in the manuscript.*
   Not entirely sure what this comment refers to. If it refers to the influence of initialization, then the answer is, that the timescales analyzed here (multiple decades away from the initialization) are not affected by any imprint of initialization anymore.

If the comment refers to the sub-sampling of 35 from the 50 members, then my reply is the following. In the setup random 35 members were picked from the full 50-member ensemble. The different subsamples have however no influence on the results, since the different 35 member subsamples share many of the same members. Hence, the different subsamples are not independent from each other, and therefore yield no substantial differences among the different subsamples. I tested this, but only very marginal differences were visible among different subsamples.

➔ **Not considered, because the comment was unclear.**

4. *The author presented well the results. I suggest that the author could make some discussion on the possible reasons of the results. For example, the strongest increases in the total risk ratio can be seen in the Scandinavian region. Why is the strongest increase in the region? Related discussion can enrich the manuscript.*
5. *The result shows that in summer the PRvar is above the PRmean, and in winter vice versa. Possible reasons should also be discussed.*
   Thank you for these two comments. The discussion will be extended by a few references and hypothesis about the reasons.
   ➔ **The discussion has been extended to cover both comments.**

**Reply to Referee comment #2**

*Wood uses a method to attribute changes in the probability of extreme events to either changing mean climate or climate variability. This is applied at a high-resolution regional scale, to explain changes to short-duration (3hr-72hr) precipitation extremes. The results are very interesting, and give an insight into the mechanisms of climate change in Europe.*

*I expect the results to be of interest to the readership of ESD. The manuscript is generally well written, though needs some clarifications and credits in places. I thus recommend acceptance after revisions.*

***General point***

*The analysis leans heavily on the calculation of probability ratio and specifically the separation in two contributing parts. This method was developed in Van der Wiel & Bintanja 2021. This study should therefore be cited in a few key locations in the manuscript, to inform the reader of the origin on the method, and to show on what background you built out with very useful insights on the drivers of changing short-duration precipitation extremes, and to give credit where it is due. I list the locations in the manuscript here, where a reference to the paper should be added:*

- *Line 75 - Add in this short summary of the paper that you will follow the Van der Weil & Bintanja (2020) methodology, rather than noting you will use 'the probability risk ratio'.*
- *Line 116 - Start the Methods section with the correct information.*
- *Line 487 - Again, in your conclusion section give credit and help the reader find the relevant reference.*
- *Figure 1 - After looking up the original paper, I noted that this figure is very close (copied from?) their figure 1. I believe you have to add "Taken after their figure 1" or*

*something, and please check if there are copyright issues (maybe not because you replotted?, I'm no expert).*

Very valid points and thank you for suggesting locations for appropriate referencing. I have checked rights and permission of the original figure, and there should be no issue. In the figure description I will place a "(adapted from van der Wiel and Bintanja (2021))".
➔ **Included the appropriate credits at the suggested locations.**

*Major points*

*Line 33 - I don't understand the addition of 'mean state' here. The magnitude of an extreme precip event is the total volume of water in a time period/for an event. Please clarify what you mean (or remove the bit between brackets).*

The information in brackets (i.e., mean state) will be removed.
➔ **Text element removed from the manuscript.**

*Figure 5 - Maybe add a note in the text on whether you have an idea on whether the trend you see, between different warming levels, is a significant one. In some regions it is steady, but e.g. in IP the lines seem to spread and come together again. I don't think there is necessarily a dynamic/circulation reason to expect this? This is useful here, but also in other places. Are the changes you note, between warming levels, extremeness, aggrevation, physically in origin?*

Some of these differences could potentially be explained by the absolute values of PRtotal. Figure 5 as well as other figures you are pointing at show the relative contributions. This means that if the PRtotal is small, then also small differences between PRvar and PRmean seem larger in the relative context. Specifically in the context of Figure 5, this could be valid for the current warming level (1°C) in several subregions (e.g., Eastern Europe, Mediterranean, Mid Europe, France). I will add one or two sentences on this in the manuscript.
➔ **Added some additional information in the manuscript.**

*Minor points*

*Line 26,27 - maybe remove one of the 'devastating' ?*

Good idea.
➔ **Removed one "devastating".**

*Line 39- 41 - Add more clearly at the beginning of the sentence that you are talking about global mean here. The next sentence came as a surprise to me.*

Will add "global" before "mean precipitation".
➔ **Clarified in the manuscript that global mean temperature is used.**

*Line 48 - The statement about changes in distribution does not at all follow from the sentence about occurrence. Maybe put the first sentence of this paragraph with the previous, and start here about the distribution/mean and variability etc.*

This is a good suggestion.
➜ **Included a brief remark in the manuscript.**

*Line 86 - 'All 50-members', add the s*

➜ **Corrected the typo.**

*Line 190 - Somewhere in this paragraph you might (if you'd like) add that there is no obvious spatial pattern in Fig 2 to be distinguished.*

Good idea.
➜ **Included a sentence on this.**

*Line 231 - What is Prudence?*

A reference will be included. PRUDENCE was a predecessor of Euro-CORDEX and defined the European subregions. These regions are widely used since in the European Regional Climate Model context and have been used in this study.
➜ **This is now clarified in the methods section.**

*Figure 4/5 - I'm not sure what AMAX is in the legend?*

AMAX stands for Annual Maximum. I will add the appropriate explanation in the figure caption
➜ **Clarified in the figure captions now.**

*Figure 4/5 - Given all your PR values are positive (i.e. above 1, or above 0), you might cut the subplots at y=1. This would give more details on the values and differences between warming levels.*

The idea was to have consistency among the different figures regarding the axis, but I see your point and will adjust figures.
➜ **Adjusted the y-axis of figures 4, 5, 7, 8, and 10 for better readability.**

*Line 346 - for the change in sign, refer to figure 7?*

Yes, that is the appropriate figure reference. I will include this.
➜ **Added the figure reference.**

*Figure 7/9 - probably too many lines to be useful/interpretable? Maybe when comparing seasons, don't show the warming levels? Then you can use the x-axis for seasons maybe.*

Visibility could, as you also suggested elsewhere, be improved by adjusting the limits of the y-Axis.
➜ **The y-axis has been adjusted for better readability.**

*Figure 10 - most of your y-axis is useless, consider cutting it off at -.4 C ?*

I will check the figures again and will adjust the y-axis extents where appropriate. For individual Figures I will however keep the axis limits homogenous throughout all subplots within.
➔ **The y-axis has been adjusted for better readability.**

*Line 423 - You note that more extreme events, have a larger variability contribution? Do you think this is due to a physical process (if so, can you hypothesise which?) or is decreased sampling here an issue?*

The latter could potentially be the case. There are grid cells where a 3-sigma event is not reached in the pre-industrial simulations and/or in the ALL simulations.
➔ **A short discussion on this has been included.**